# The Feasibility and Applicability of Stem Cell Therapy for the Cure of Type 1 Diabetes

**DOI:** 10.3390/cells10071589

**Published:** 2021-06-24

**Authors:** Ryota Inoue, Kuniyuki Nishiyama, Jinghe Li, Daisuke Miyashita, Masato Ono, Yasuo Terauchi, Jun Shirakawa

**Affiliations:** 1Laboratory of Diabetes and Metabolic Disorders, Institute for Molecular and Cellular Regulation (IMCR), Gunma University, Maebashi 371-8512, Japan; rinoue@gunma-u.ac.jp (R.I.); m2120e02@gunma-u.ac.jp (K.N.); jinghe_li@gunma-u.ac.jp (J.L.); 2Department of Endocrinology and Metabolism, Graduate School of Medicine, Yokohama City University, Yokohama 236-0004, Japan; daisuke1535@yahoo.co.jp (D.M.); m.731.ono@gmail.com (M.O.); terauchi@yokohama-cu.ac.jp (Y.T.)

**Keywords:** diabetes, type 1 diabetes, stem cell therapy, pancreatic β-cell

## Abstract

Stem cell therapy using islet-like insulin-producing cells derived from human pluripotent stem cells has the potential to allow patients with type 1 diabetes to withdraw from insulin therapy. However, several issues exist regarding the use of stem cell therapy to treat type 1 diabetes. In this review, we will focus on the following topics: (1) autoimmune responses during the autologous transplantation of stem cell-derived islet cells, (2) a comparison of stem cell therapy with insulin injection therapy, (3) the impact of the islet microenvironment on stem cell-derived islet cells, and (4) the cost-effectiveness of stem cell-derived islet cell transplantation. Based on these various viewpoints, we will discuss what is required to perform stem cell therapy for patients with type 1 diabetes.

## 1. Introduction

The destruction of insulin-producing β-cells is the main pathophysiological feature of type 1 diabetes (T1D) [1,2]. Blood glucose levels can be reduced by daily multiple exogenous insulin injections or continuous insulin infusion via a pump. However, because the exogenous insulin does not pass through the liver, the insulin is unable to regulate hepatic glucose levels, resulting in unstable glycaemic control in T1D patients [3]. Insufficient glycaemic control induces long-term complications in T1D patients. Sensor augmented insulin-pump therapy and transplantation therapy of whole pancreas or pancreatic islets may be useful strategies to compensate for β-cell function and control the blood glucose levels [4]. Although pancreas transplantation provides a high rate of normalization of the blood glucose levels and insulin withdrawal, it requires a technically difficult surgery and can have several postoperative complications, including portal vein thrombosis [5]. Pancreas transplantation also improves the quality of life; however, the number of donors is limited, and immunosuppression therapy is required [6]. Reportedly, patients with T1D were able to withdraw from insulin therapy after undergoing islet transplantation from brain-dead donors accompanied by steroid-free immunosuppressive therapy [7]. Islet transplantation reportedly reduced hypoglycaemic events and improved glycaemic control in T1D patients, compared with insulin injection therapy [8]. Islet transplantation has a lower risk of surgical complications than pancreas transplantation, but repeated transplantations are often required to enable withdrawal from insulin therapy [9]. Thus, the transplantation of both pancreas and islets has problems associated with donor shortage, immunosuppressive therapy, and graft rejection. As a method that can cope with the shortage of donors, a therapeutic strategy for producing pancreatic islet-like cells from human pluripotent stem cells (hPSCs) including induced pluripotent stem cells (iPSCs) or embryonic stem cells (ESCs) and applying them to transplantation can be considered. In the past 15 years, basic methods for inducing the differentiation of human stem cells towards pancreatic endocrine cells, generated from the formation of definitive endoderm to pancreatic endoderm during embryogenesis, followed by differentiation into pancreatic progenitors have been established, enabling a yield of 20–40% insulin^+^ pancreatic islet-like cells [10,11,12,13,14,15]. Because islet cells have a three-dimensional (3D) structure and interact with surrounding cells in vivo, “co-culture” models, such as the co-culture of islet cells and endothelial cells, should be considered [16,17,18]. In vitro, several methods have been used to generate engineered pancreatic islets with a 3D structure, known as “pseudo-islets”, which are composed of α, β, δ, PP and ε cells [19,20]. Micallef et al. generated a human embryonic stem cell reporter that encodes green fluorescent protein (GFP) at the INS locus to facilitate the characterization of insulin-producing cells [21]. Figure 1 shows the images of human pluripotent stem cells (hPSCs)-derived islet organoids. The transplantation of stem cell-derived pseudo-islets has the potential to improve insulin secretion in diabetic rodents [22]. However, how the immune response and the changes in β-cell function progress when stem cell-derived islet cells are transplanted remain largely unclear. Moreover, the cost-effectiveness of stem cell therapy for β-cells or pseudo-islets, compared with conventional diabetes therapy, must also be considered.

In this review, we will focus on the issues that should be considered before performing stem cell therapy for T1D patients, such as the autoimmune response, a comparison with traditional insulin therapy, the microenvironment, and the cost.

## 2. Autoimmune Responses in Stem Cell Therapy for T1D

The autologous transplantation of hPSCs-derived pancreatic islet-like cells has the potential to provide glycaemic control without immune rejection. It is thought that autologous transplantation of iPSCs does not cause graft rejection because those iPSCs are derived from patients themselves. In fact, autologous transplantation of mouse iPSCs-derived skin, bone marrow, endothelial cell, or neuronal cells and monkey iPSCs-derived neuron did not evoke immune responses [23,24,25]. Meanwhile, Zhao et al. reported that the autologous transplantation of mouse iPSCs-derived teratoma into subcutaneous space showed graft rejection [26]. This report indicated that the abnormal gene expression during the differentiation of iPSCs could induce a T-cell-dependent immune response in autologous transplantation. An autoimmune response by autoreactive T cells for transplanted islet cells can occur in T1D recipients. In this chapter, we will review the autoimmune responses that can occur during the transplantation of stem cell-derived islet-like cells (SC-islet).

The transplantation of pancreatic islets derived from hPSCs ameliorated hyperglycaemia in diabetic mice [27]. Consequently, the autologous transplantation of iPSCs is considered to be useful, but a huge amount of time and cost is required to culture the cells and to promote their differentiation into insulin-producing β-cells. Therefore, banks of iPSCs corresponding to various HLA are being constructed for allogeneic transplantation [28]. An encapsulation device is also useful for T1D patients to prevent autoimmune reactions against transplanted SC-islet [29]. As hPSCs carry a risk of tumorigenesis, such as teratoma, these encapsulation devices might be better transplanted subcutaneously. It has been reported that the incomplete reprogramming of somatic cells induces tumorigenesis because of changes in epigenetic regulation [30,31]. The pharmacological inhibition of lysine-specific demethylase 1 (LSD1), a histone demethylase, has been reported to prevent teratoma formation from iPSCs transplanted into immunodeficient mice [32].

Human leukocyte antigen (HLA) distinguishes foreign antigens and elicits immune responses [33,34]. HLA class II is expressed on antigen-presenting cells such as dendritic cells, and present the antigen to helper T cells, resulting in the initiation of antigen-specific immune responses. The class II gene shows the strongest association with T1D, and the haplotype formed by the DR and DQ genes is involved in the disease susceptibility of T1D [35,36]. Transplantation therapy using mesenchymal stem cells (MSCs) has also been reported to be useful because MSCs lack the expression of HLA class II antigens [37]. The transplantation of MSC-derived insulin-producing islet-like cells improved glycaemic control in diabetic STZ-treated mice despite the infiltration of immune cells into the peritoneal cavity and left kidney capsule after local transplantation [38]. A clinical trial showed that MSC transplantation improved glycaemic control in T1D patients [39]. On the other hand, no clinical trials have examined the use of SC-islet cells from iPSCs, ESCs or MSCs to date.

The prevention of graft rejection for SC-islet cell transplants is an important issue. Yamaguchi et al. reported the autologous transplantation study where mouse-PSC-derived islet cells were transplanted into the kidney capsule of mice with STZ-induced diabetes [40]. In that report, SC-islets were generated by injecting mouse PSCs into Pdx-1-deficient rat blastocysts, and the SC-islets contained endothelial cells from rat origin. As a result, immunosuppressive therapy was required for the first 5 days after transplantation. Even after the withdrawal of immunosuppressive drugs, SC-islets continuously improved blood glucose levels within the normal range in diabetic mice for 370 days. However, whether the autologous transplantation of iPSCs-derived islets from patients with T1D can cause graft rejection remains unclear. Leite et al. conducted an in vitro experiment in which SC-islet cells from T1D subjects or non-diabetes subjects were co-cultured with autologous immune cells, and they reported that endoplasmic reticulum stress caused an immune response in SC-islet cells from both T1D donors and non-diabetes subjects, indicating that immune responses can also occur with autologous transplants [41]. This report also showed that T cell activation is restricted to the autologous transplant of β-cell-enriched SC-islet cells and does not occur in α-cell-enriched SC-islet cells; however, the mechanism of β-cell-specific T cell activation is not fully understood.

Considering the treatment of stem cell therapy for T1D patients, it is important to clarify how SC-islet cells from T1D donors differ from SC-islet cells from non-diabetes subjects. Millman et al. reported that there were no differences between T1D and non-diabetes SC-islet cells in response to cytokine-induced stress, such as changes in interleukin-1β (IL-1β), tumour necrosis factor-α (TNF-α), or interferon-γ (INF-γ) [42]. SC-islet cells from T1D and type 2 diabetes (T2D) patients reportedly showed insulin secretion comparable to that of SC-islet cells from non-diabetic subjects [43]. On the other hand, Hosokawa et al. reported that when SC-islet cells from patients with fulminant T1D and healthy subjects were treated with inflammatory cytokines (TNF-α, IL-1β and IFN-γ), apoptosis was more likely to occur in the SC-islet cells from the fulminant T1D subjects [44]. Immune response-related genes are differentially expressed in SC-islet cells from fulminant T1D donors, compared with those from control subjects, suggesting that abnormal immunoregulation in the fulminant T1D β-cells might cause rapid β-cell destruction and disease development. Therefore, any conclusions regarding the features of SC-islet cells from T1D subjects are difficult to make, at present. Even during the autologous transplantation of SC-islet cells from T1D patients, the possibility of graft rejection should be considered.

Gene modification approaches might be useful for the protection of SC-islet cells from immune responses. The deletion of HLA-A in hematopoietic stem cells improved both engraftment and haematopoiesis in immunocompromised mice [45]. Immunomodulatory proteins, such as programmed death-ligand 1 (PD-L1) and cytotoxic T-lymphocyte-associated protein 4 (CTLA4), can be therapeutic targets for T1D. Inhibitors of these proteins, which are known as immune checkpoint inhibitors, exert an antitumor effect by activating T cells. However, treatment with immune checkpoint inhibitors can reportedly lead to the development of T1D as an adverse effect [46]. The overexpression of PD-L1 and CTLA4Ig by the adeno-associated virus in mouse pancreatic β-cells preserved the β-cell mass and protected NOD mice from T1D development [47].

An approach that focuses on immune cells might be useful for the treatment of autoimmune disease. A recent report indicated that mitochondrial reactive oxygen species (mtROS) were increased in regulatory T cells (Tregs) in experimental autoimmune encephalitis (EAE) mice, which is a model of autoimmunity disease that exhibits abnormal Tregs function [48]. In EAE mice, scavenging mtROS in Tregs diminished the autoimmune responses [48]. Joshi et al. reported that the macrophages which differentiated from T1D-derived iPSCs specifically presented a pro-insulin peptide to islet-infiltrating T cells isolated from that same donor, leading to T cell activation [49]. This T cell activation was specifically blocked by anti-HLA-DQ antibodies.

Based on these reports, research focusing on not only pancreatic β-cells but also the immune cells of T1D patients will be very important for promoting treatment strategies for diabetes using iPSCs (Figure 2).

## 3. Comparison of Stem Cell Therapy for Pancreatic β-Cells with Automated Insulin Delivery System

T1D is characterized by chronic hyperglycaemia caused by insulin deficiency arising from the destruction of pancreatic β-cells, mainly by autoimmune mechanisms, thereby requiring life-long insulin injection [2]. At present, insulin pumps help improve the management of blood glucose levels. The concept of time in range (TIR) has been proposed as a target range for glycaemic control, and the recommended range for general adults with T1D or T2D is a blood glucose level within 70–180 mg/dL for at least 70% of the day [50]. Blood glucose monitoring using continuous glucose monitoring (CGM) or flush glucose monitoring (FGM) and insulin pumps with a Predictive Low Glucose Suspend (PLGS) function has helped to better maintain blood glucose levels [51,52]. Sensor augmented pump (SAP) therapy is a combination of CGM and an insulin pump and is considered to be an artificial pancreas, since the amount of insulin can be adjusted based on the CGM value [53]. In addition, β-cell replacement therapy is also expected to be a promising treatment for T1D to realize the goal of diabetes therapy that is free from insulin injection. β-cell replacement includes transplantation of pancreas, islets or SC-islet cells. In this chapter, we would like to compare the transplantation of SC-islet cells and artificial pancreas (Table 1).

### 3.1. Stem Cell Therapy

The two main advantages of stem cell therapy for patients with T1D are that insulin injections and blood glucose monitoring are not required and that the TIR can be improved. Under 20 mM high glucose stimulation, iPSCs-derived islets generated from T1D patients showed insulin secretion similar to that of iPSCs-derived islets generated from non-diabetic patients, indicating glucose-dependent insulin secretion (2.0 ± 0.4 vs. 1.9 ± 0.3 mIU/10^3^ cells) [42]. Kim et al. generated iPSCs-derived insulin-secreting cells from T1D and T2D subjects and compared them with those from healthy controls, and each derived cell secreted insulin dependent on the blood glucose levels [43]. It is a physical and psychological burden for T1D patients to have to measure their blood glucose level and inject insulin every time they eat, but stem cell therapy can free T1D patients from blood glucose monitoring and insulin injection. There is also no significant difference in insulin secretion between T1D-derived iPSCs and those from healthy subjects. Therefore, stem cell therapy for β-cells can potentially improve glucose-stimulated insulin secretion, resulting in better control of the TIR and a reduction in the frequency of hypoglycaemia.

There are three main disadvantages of stem cell therapy for patients with T1D. The first is the immune response to transplantation. There are two types of transplantation of pancreatic islet cells differentiated from iPSCs: allogeneic transplantation and autologous transplantation. The key barrier for both types of transplantation is the control of the immune response. However, autologous transplantation of iPSCs might be useful for avoiding rejection because they are not thought to initiate immune responses [23,24].

The second disadvantage is that there is a potential for the transplanted cells to become cancerous. The c-Myc gene, one of the Yamanaka factors, is known to cause cancer in iPSCs, but oncogenesis can be prevented by using L-Myc instead of the c-Myc gene [28]. In addition, the presence of undifferentiated cells is also a cause of cancer. One report showed that orlistat, an anti-obesity drug, selectively removed undifferentiated hPSCs by inhibiting fatty acid synthesis [54].

The third disadvantage is the possibility that insulin secretion of the transplanted cells may be insufficient. iPSCs-derived islet cells transplanted into mice reportedly secreted insulin for several months or more after transplantation [55]. However, long-term studies are required to investigate whether the insulin secretion capacity is sufficient for clinical application. If transplanted iPSCs were to become afunctional or cancerous, they would need to be resected from the body.

### 3.2. Artificial Pancreas

The advantage of the artificial pancreas is that it improves TIR and reduces the frequency of hypoglycaemia without requiring immunosuppression. After tissue or cell transplantation, the administration of immunosuppressive drugs is necessary to prevent immune rejection, and this can be accompanied by adverse effects such as susceptibility to infection, nephrotoxicity, and myelosuppression. An artificial pancreas can improve the TIR to the same extent as an endogenous pancreas. SAP therapy combined with an insulin pump and CGM is widely used in clinical practice for the treatment of T1D. For better glycaemic control, a closed-loop system is being developed to adjust the insulin infusion rate using an analysis algorithm based on CGM. According to Thabit et al., the percentage of time that glucose levels were within the target range was 11.0 percentage points higher in the closed-loop system group than in the SAP group (*p* < 0.001), and the mean glucose and HbA1c levels were significantly lower in the closed-loop system group than in the SAP group *(p* < 0.001, *p* = 0.002, respectively). The area under the curve (AUC) for glucose levels below 63 mg/dL was 39% lower in the closed-loop system group (*p* < 0.001) [56].

There are four disadvantages to the use of an artificial pancreas. The first disadvantage is the complex maintenance that is required. For example, SAP therapy, which is one type of artificial pancreas, requires the regular replacement of sensors for CGM and insulin infusion sets. Second, there is a risk of hypoglycaemia and diabetic ketoacidosis (DKA). An artificial pancreas requires that T1D patients adjust the insulin that is administered according to their diet, and there is a risk of hypoglycaemia if the patients accidentally administer more insulin than necessary. Other problems with an insulin pump include equipment trouble such as the occlusion of the insulin cannula, which can interrupt insulin infusion resulting in acute hyperglycaemic complications, such as DKA and hyperosmolar hyperglycaemic syndrome (HHS). A third disadvantage is the occurrence of local skin problems, such as lipoatrophy, lipohypertrophy, eczema and wounds. According to Berg et al., skin problems add to the cost of medical care for both paediatric ($80 per person) and adult ($40 per person) patients [57,58].

The advent of the artificial pancreas has brought many benefits to T1D patients, including improved TIR and less frequent hypoglycaemia, but it has yet to eliminate the need for insulin injection and blood glucose monitoring.

## 4. Islet Microenvironment and SC-Islet Cells

The addition of specific factors at the correct differentiation stages results in producing functional islets from hPSCs which secrete insulin in response to glucose [59,60,61]. Those SC-islets, however, had a lower insulin secretion capacity, compared with native human islets, and SC-islets were still less matured (for example, they lacked the expressions of urocortin3, MAF A, and SIX3) [62]. The results of an ongoing clinical trial of SC-islet transplantation for patients with T1D (NCT02239354) will provide us with more information about the maintenance of graft functions [63,64]. It was reported that the islet microenvironment and interactions among pancreatic endocrine cells played crucial roles in β-cell maturation, function, and survival [65]. The islet microenvironment (also referred to as the islet niche) consists of endocrine cells (α-cells, δ-cells, PP-cells, or ε-cells), pancreatic acinar cells, ductal cells, mesenchymal stromal cells, endothelial cells, resident macrophages, and extracellular matrices. Additionally, abundant islet vascular networks and interactions between islets and sympathetic or parasympathetic nerves also exist. This microenvironment is a prerequisite for revascularization of the grafted islets; however, the underlying mechanism remains unclear. A direct differentiation protocol from iPSCs produces not only β-cells, but also α-cells (expressing glucagon), δ-like cells (expressing somatostatin), and non-endocrine cells (which are similar to acinar cells, ductal cells, mesenchymal cells and entero-chromaffin cells) [62]. Here, we will focus on the islet microenvironment required for successful stem cell therapy for T1D.

### 4.1. Endocrine Cells

Human islets consist of several endocrine cells including α-cells, β-cells, δ-cells, PP-cells and ε-cells. Here, we will review the interactions among α-, δ- and β-cells [66,67,68]. β-cells express glucagon receptor and somatostatin receptor on their plasma membrane, and cross-talk among α-, β- and δ-cells occurs in a paracrine manner. α-cells secrete glucagon, which enhances GSIS from β-cells and promotes β-cell survival and proliferation [69]. α-cells also reportedly secrete acetylcholine, which affects the muscarinic acetylcholine 3 receptors in β-cells and stimulate insulin secretion. β-cells and δ-cells are directly connected by gap junctions, and Ca^2+^ influx and insulin secretion from β-cells are orchestrated by δ-cells. Urocortin-3, which acts on δ-cells, is also secreted from β-cells together with insulin, thereby stimulating somatostatin secretion [70]. Acetylcholine secreted from α-cells also enhances the secretion of somatostatin from δ-cells, which in turn inhibits both insulin and glucagon secretion in humans [71]. These paracrine interactions among endocrine cells in human islets play important roles in the fine-tuning of insulin secretion in a manner appropriate to the situation. On the other hand, whether the interactions among endocrine cells derived from iPSCs affect β-cell function or survival remains unknown.

SC-islet cells lack urocortin In vitro, and so one of the interactions between β- and δ-cells may be diminished [62]. This may cause the insufficient maturation of SC-islet cells. Whether α-cells derived from stem cells secrete acetylcholine or not is also unknown. Furthermore, the ratio of the number of α-, β- and δ-cells and the orientations and direct connections of each cell type in SC-islets remains unclear.

### 4.2. Exocrine Cells

When islets are isolated, a certain number of acinar cells are attached to the islets. The destruction of these acinar cells during isolation procedures and culturing reportedly has negative effects on islet survival [72]. Dying acinar cells secrete proteolytic enzymes, such as trypsin and chymotrypsin, which harm islets insulin granules and decrease the islet mass. When acinar cells are transplanted together with islets, they may provoke an inflammatory response and disturb the revascularization of the graft [73]. On the other hand, the effect of ductal cells on islet transplantation is controversial. The co-transplantation of human islets and ductal cells decreased insulin secretion from islets, compared with the transplantation of pancreatic islets alone, indicating that the β-cells of islets with ductal cells are less mature than native human islets [74]. Ductal cells secrete pro-inflammatory mediators such as IL-1β, TNF-α, CD40, nitric oxide and IGF-II, and they may decrease the β-cell mass in transplanted islets [75]. However, ductal cells also secrete IL-8 and vascular endothelial growth factor (VEGF), which promote graft revascularization and improve graft survival [76]. Whether pancreatic exocrine cells are indispensable for SC-islet cell function and survival remains unclear. Recently, an in vivo study showed the presence of bidirectional blood flow between islets and exocrine tissues, and this endocrine-to-exocrine interaction seemed to be related to islet function and survival [77]. Interactions among transplanted SC-islet cells and pancreatic exocrine cells of the recipient have not been previously reported. A strategy that focuses on the interactions between exocrine cells and SC-islet cells may be beneficial.

### 4.3. Mesenchymal Stromal Cells, Endothelial Cells, Nerves, and Macrophages

In the pancreas, mesenchymal stromal cells (MSCs), endothelial cells, autonomic nerves, and resident macrophages also exist as well as endocrine and exocrine cells. Pancreatic MSCs have anti-inflammatory effects and proangiogenic effects on β-cells [78]. When MSCs from the human pancreas are cultured with dimethyloxallyl glycine and TNF-α, nuclear factor erythroid 2-related factor 2 (NRF2) and VEGF are upregulated; these factors may promote graft revascularization and graft survival. Endothelial cells also act effectively on β-cells maturation and survival [79]. Native β-cells exhibit polarity, with an apical and basolateral membrane and a greater density of insulin vesicles in the basal region, close to the endothelial cells. Islets secrete VEGF-A, which promotes endothelial cell development and intra-islet capillaries. Endothelial cells are related to the formation of the vascular basement membrane, and SC-islet cells exhibit polarity when co-cultured with endothelial cells, indicating β-cell maturation [80]. β-cells co-cultured with endothelial cells also showed longer survival. In addition to revascularization, the interactions between islets and autonomic nerves are also important for the maintenance of appropriate β-cell function. The activation of sympathetic nerves inhibits insulin secretion, while parasympathetic activation stimulates insulin secretion from β-cells. Both sympathetic and parasympathetic nerves stimulate glucagon secretion from α-cells. Human islets are sparsely connected by autonomic axons, and sympathetic nerves innervate the smooth muscles of blood vessels within islets [81]. On the other hand, parasympathetic nerves slightly penetrate human islets, so autonomic nerves may regulate hormone secretion in human islets by constricting or diluting the microvessels and controlling intra-islet blood flow. Activated macrophages, which show an inflammatory phenotype (M1 macrophages), may cause graft rejection during islet transplantation. On the other hand, resident macrophages shift to an anti-inflammatory phenotype (M2 macrophages) during β-cell regeneration [82]. Additionally, macrophages secrete VEGF-A, which promotes islet revascularization and β-cell regeneration [83]. Since the ablation of macrophages diminishes islet revascularization or β-cell regeneration, macrophages may be essential for β-cell survival. When SC-islet cells are transplanted, recipient MSCs and endothelial cells may change in number or function, although this has not yet been elucidated. Interaction between transplanted SC-islets and resident macrophages may differ by the transplant site [84]. Furthermore, the effects of macrophages on SC-islet cells may be different from those of mouse islets or human islets. Further understanding of these components may be useful for enhancing graft survival or improving the functions of SC-islet cells.

### 4.4. Extracellular Matrices

Extracellular matrices (ECMs) consist of several cytoskeletal components, such as collagen, laminin, vitronectin, fibronectin proteoglycans, and matricellular proteins, as well as growth factors, such as connective tissue growth factor (CTGF), platelet-derived growth factor (PDGF), VEGF, fibroblast growth factor (FGF), hepatocyte growth factor (HGF), and insulin-like growth factor (IGF). ECMs provide an anchor, creating a scaffold to stabilize islets and other components, modulate second messenger signalling, and promote β-cell proliferation and survival and protect against apoptosis [65,80]. The proteins of the basement membrane (BM), a thin layer of the ECMs, are required for β-cell polarization. Islets have two types of BM: peri-islet BM, and vascular BM. The peri-islet BM separates exocrine compartments from the islet and acts as a barrier against infiltrating leukocytes. The vascular BM is associated with islet capillaries and endothelial cells [85]. Native β-cells have three domains, the apical, lateral and basal domains, and express zonula occludens proteins 1 (ZO-1) at their apical domain, E-cadherin at their lateral domain, and liprin at their basal domain [85]. SC-islet cells cultured without BM proteins in vitro showed incorrect polarization and an inappropriate expression of ZO-1, liprin, and E-cadherin within the same domain; these SC-islet cells secreted less insulin, compared with native β-cells. In the presence of BM proteins, SC-islet cells are in contact with the BM at the capillary interface, expressing liprin within this domain and expressing ZO-1 in the opposite domain. These polarized β-cells can control proper insulin secretion [86].

The connection between nerves and vessels in islets requires ECMs to support vascularization around each islet. In the present islet isolation protocol, SC-islet cells and other cells are dissociated, and only SC-islet cells are purified, then re-aggregated. Much of the microenvironment is therefore lost in these cell-purification or decellularization processes. Hydrogel scaffolds made from decellularized ECMs components can be used to replicate the effects of ECMs on β-cells. When SC-islet cells are cultured in a hydrogel, the proliferation rate is increased, and the apoptosis rate is decreased compared with SC-islet cells cultured without a hydrogel [87].

### 4.5. Transplantation Site

Grafts cannot be transplanted within the pancreas because of the abundance of digestive enzymes, and if the transplanted cells were to form a teratoma, they would be difficult to remove. Unlike other stem cell therapies under investigation (neurons, retina, skin tissue, cardiomyocytes, and hematopoietic cells), cadaveric islets or SC-islet cells are transplanted at a site where native islets or β-cells do not exist (e.g., portal vein, subcutaneously). Adjusting the site to promote engraftment prior to transplantation might be advantageous, in addition to refining the transplantation of islets or SC-islet cells.

In a mouse study, before transplantation, the insertion of scaffolds loaded with VEGF and fibrinogen promoted prevascularization of the planned implantation site within the epididymal fat pad [88]. However, this strategy was not useful at other sites, such as a subcutaneous space or the small bowel mesentery. Which sites are suitable for prevascularization in humans remain unknown. Endothelial cells may be beneficial for vascularization, but they might also be vulnerable to high glucose levels, leading to cell death and dysfunction [89]. Therefore, it may be important to maintain a normal glucose level in the recipient to maintain the function of endothelial cells prior to transplantation.

The islet microenvironment is associated with β-cell function and survival. Some components are preferable for the promotion of the functional maturation and survival of β-cells, such as MSCs and endothelial cells, but little is known about other cells or components of the microenvironment (Figure 3).

## 5. Cost-Effectiveness of Stem Cell-Derived Islet Cells Transplantation Therapy for T1D

There are more than 463 million people with diabetes worldwide, and the healthcare costs associated with the treatment of complications and the improvement of hyperglycaemia are huge. The treatment of T1D involves insulin intensive therapy (IIT), insulin pump therapy, pancreatic transplantation, and islet transplantation. Pancreas or islet transplantations are enabled to regulate the blood glucose levels in T1D patients with unstable blood glucose levels [90,91]. Importantly, these treatments not only improve a patient’s quality of life but can also lead to a reduction in lifetime healthcare costs. On the other hand, pancreas or islet transplantations has some disadvantages, such as a lack of donors for cadaveric pancreas and islets and the need to administer immunosuppressive drugs. The transplantation of SC-islet cells could alleviate this donor deficiency; however, the production of pancreatic β-cells is very expensive, possibly preventing the reductions in lifetime healthcare costs that are associated with SC-islet cells. This chapter will provide the latest findings focusing on the cost of SC-islet cells transplantation (Table 2).

T1D, an insulin-dependent condition, costs $133.7 billion in healthcare costs per year, and the total income loss has been calculated at $289.2 billion [92,93]. From another perspective, in 2017, patients using insulin in the U.S. paid $9601 per year in healthcare costs [94]. Patients with T1D are also at a high risk of related complications, such as amputation, blindness, and renal failure, because of their unstable glycaemic control [93]. If transplants enable autonomous blood glucose control, these procedures could prevent diabetes-related complications and reduce lifetime healthcare costs [95].

Islet transplantation has been approved in several countries [96,97,98]. Islet transplantation is estimated to cost $13,872. In 2012, Cooper-Jones, Ford et al. reported a cost-effectiveness study suggesting that islet transplantation would be more cost-effective than insulin therapy for high-risk T1D patients after 9 to 10 years [99]. In addition, islet transplantation is estimated to cost $519,000 per patient over 20 years, whereas insulin therapy is estimated to cost $663,000 per patient over 20 years [99]. However, this study assumed a single transplant over a lifetime, which is likely to be an underestimation of the actual treatment. On the other hand, Wallner et al. reported the cost-effectiveness of patients with up to four islet transplantation procedures from the perspective of Canadian insurers [99]. According to this study, islet transplantation increased the average lifespan by 3.3 years, but the cost of transplantation was so high that cost-effectiveness far exceeded the willingness-to-pay criteria of Canadian insurers [100]. Although a few reports have discussed the cost of cadaveric islet transplantation, it is important to reduce the number of transplants and the cost of immunosuppressive drugs.

If immunosuppressive drugs were not required, the healthcare costs of islet transplantation would be reduced. One way to avoid the administration of immunosuppressive drugs is through the autologous transplantation of human iPSCs-derived islet cells. A cost-effectiveness analysis of this transplantation reported that cost-effectiveness was achieved by preventing complications for 8 years after transplantation, compared with IIT [100]. However, this report did not sufficiently consider the economic aspects of the bioprocess. Another way to avoid autoimmunity has been to encapsulate SC-islet cells and transplant them into patients [100]. The encapsulation of SC-islet cells prior to transplantation allows them to detect blood glucose levels and secrete insulin while remaining isolated from the immune system of the recipient [101]. Bandeiras et al. provided a cost-effectiveness analysis of encapsulated SC-islet cell transplantation that considered the cost of SC-islet cell production [102]. In their paper, the operational efficiency of the machine, labour, equipment, and reagent costs were taken into account assuming a demand of 50 patients per year, and the simulation resulted in a manufacturing cost per patient of about $80,000, which was higher than that for cadaveric islets [103].

As the annual demand increases, the cost of labour and facilities will decrease. However, the cost of reagents, especially those related to the differentiation stage, will continue to be relatively large. The results of a bioprocess model, which assumed a strategy of manufacturing products for 50 patients per year and 10 patients per machine, also assumed a total final cost of $650,000 per patient for the transplantation [102]. This model estimated that, compared with IIT, SC-islet cells transplantation would increase the quality-adjusted life years (QALYs) in 96.4% of the patients by an average of 3.73 years per patient over a 20-year patient period. However, the healthcare cost of SC-islet cells transplantation over 20 years is four times the cost of IIT.

Cost-effectiveness is measured by the additional cost of gaining 1 QALY compared to the existing treatment. The incremental cost-effectiveness ratio (ICER) is a measurement of cost-effectiveness. Whether insurers in each country are likely to be persuaded by the new treatment depends on the ICER, which indicates the cost-effectiveness, and the willingness to pay (WTP) threshold. In the UK healthcare system, as a guide to the cost-effectiveness of new treatments, implementation is recommended at an ICER threshold of £20,000 ($26,089) [100]. Furthermore, the American Diabetes Association recommends an ICER of $50,000 for most treatments [104]. At these thresholds and using the above-described costs, islet cell transplantation is only cost-effective for as many as 2% of patients. This suggests that the costs associated with transplantation must be drastically reduced for this treatment to be considered more positively than insulin therapy.

Additionally, when determining the cost-effectiveness of SC-islet cells transplantation, the clinical efficacy of cadaveric islet transplantation is often premised, which is a limitation in calculating accurate cost-effectiveness. However, even if SC-islet cells transplantation is as effective as cadaveric islet transplantation for T1D, lowering the manufacturing cost remains an important task.

The most important factors for cost reduction are the scale-up of manufacturing capability and supply chain management. The cost of generating SC-islet cells could be reduced if embryonic stem cells could be used [105]. However, because of ethical limitations, somatic cells must be reprogramed into pluripotent stem cells. In terms of technology and patents, reducing the cost of reprogramming somatic cells into pluripotent stem cells is likely to be difficult. Thus, low-cost and more efficient reprogramming methods for somatic cells are also required.

In conclusion, SC-islet transplantation for the treatment of diabetes increases QALYs and prevents complications, although a reduction in manufacturing costs will be essential to achieve cost-effectiveness (Table 2). To reduce manufacturing costs, manufacturing and supply chain management must be scaled up.

## 6. Discussion

Stem cell therapy is considered to be an attractive treatment strategy with the potential to enable the withdrawal of insulin therapy in patients with T1D. Furthermore, autologous transplantation using iPSCs-derived islet cells is expected to have the potential to treat T1D patients without requiring immunosuppressive therapy. At the present time, however, it is difficult to prepare a sufficient amount of iPSCs-derived autologous islet cells because of the high cost and technical problems. In the future, technology capable of producing a sufficient amount of SC-islet at a lower cost will be required. On the other hand, an immune response can sometimes occur even if autologous transplantation is performed. The pathophysiology of T1D is not yet fully understood, partly because a suitable animal model for studying T1D does not exist. Experiments using SC-islet from patients with T1D may help researchers to understand the pathophysiology of T1D. Investigating the characteristics and behaviour of iPSCs-derived immune cells (e.g., macrophages, T cells) in patients with T1D as well as the characteristics of β-cells will also be important.

In mouse islets, interactions among β-cells and with other pancreatic endocrine cells, endothelial cells, the extracellular matrix and autonomic nerves reportedly affect β-cell function and survival. Thus, generating organoids of islets or pancreas from hPSCs and transplanting them into T1D patients might also be an important therapeutic strategy [22,106]. With each approach, however, further research is required because of cost issues. In summary, research using iPSCs is expected to be useful for advancing the treatment of T1D.

## Figures and Tables

**Figure 1 cells-10-01589-f001:**
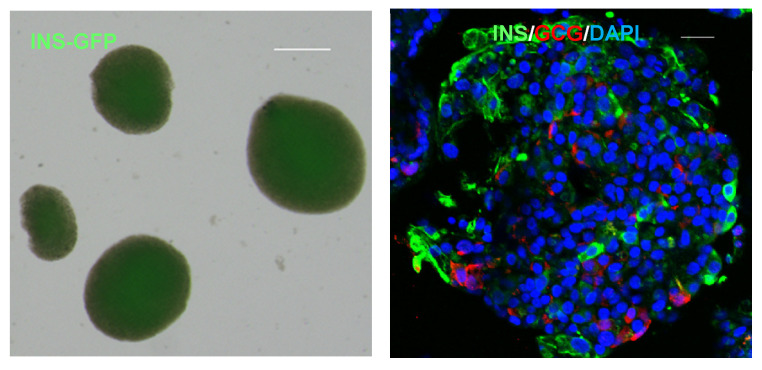
Human pluripotent stem cells (hPSCs)-derived pancreatic islets. Live-cell imaging (**left**) and immunofluorescence (**right**) of hPSCs-derived pancreatic islet organoids. GFP is expressed under the insulin promoter. Cells are stained with antibodies to insulin (green), glucagon (red) and DAPI (blue). The scale bars in both panels represent 20 μm.

**Figure 2 cells-10-01589-f002:**
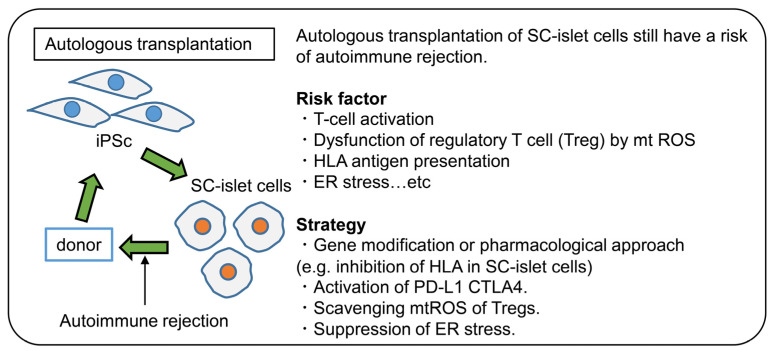
Autologous transplantation of SC-islet in T1D. Even the autologous transplantation of SC-islet has the potential for rejection. Thus, there is a need to further understand the pathophysiology of the immune response in T1D. Furthermore, strategies for avoiding the immune response are required.

**Figure 3 cells-10-01589-f003:**
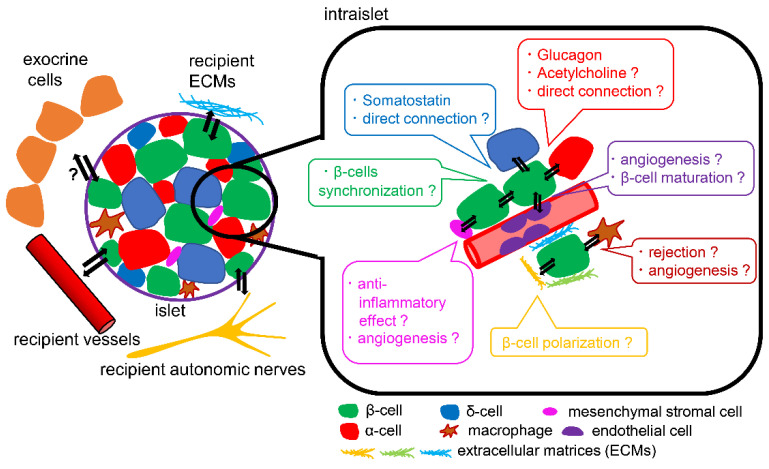
Schema of islet microenvironment. The interactions of β-cells with α-cells, δ-cells, endothelial cells, mesenchymal stromal cells, macrophages, and ECMs can affect β-cell function and survival. The differences in the interactions with microenvironments between transplanted hPSCs-derived islets and transplanted human islets remain unclear.

**Table 1 cells-10-01589-t001:** Comparison between SC-islet cells transplantation and automated insulin delivery systems.

	SC-Islet Cells Transplantation	Automated Insulin Delivery Systems
**Advantages**	Free from insulin injection	No immunosuppression
	Improved time-in-range	Improved time-in-range
**Disadvantages**	Risk of immunosuppression	Complicated maintenance
	Risk of cancerization	Adjustment of dose according to diet
	Risk of insulin insufficiency	Risk of hypoglycemia, DKA or HHS
	Risk of re-transplantation	Local skin troubles

**Table 2 cells-10-01589-t002:** Reported costs of pancreatic β-cell transplantation using cadaveric islets or stem cells.

Source/Country	Study Population	Time Horizon	Intervention and Comparator	Per Additional QALY	Results
Wallner et al., 2018 (Canada)	Average age of 47 years, adults with hypoglycemia unawareness	62.5 years	Intervention: Cadaveric islet cell transplantation Comparator: Intensive insulin therapy	$150,006	Islet cell transplantation had a probability of being cost-effective of 0.5% at a willingness-to-pay (WTP) threshold of $100,000 per QALY.
Banderias et al., 2019 (USA)	Age of 18–35 years, hypoglycemia unawareness	20 years	Intervention: Stem cell-derived islet cell transplantation Comparator: Intensive insulin therapy	$249,740	Islet cell transplantation was cost-effective for 3.4% of patients when a WTP of up to $150,000 per QALY was considered.

## Data Availability

Not applicable.

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
