# Peer review of "The Feasibility and Applicability of Stem Cell Therapy for the Cure of Type 1 Diabetes"

_cells, 2021, doi:10.3390/cells10071589_

Round 1

Reviewer 1 Report

  1. The authors must be aware of the fact that when they use the term "pseudoislet" it should refer to a cell cluster comprised of α, β, δ, PP and ε cells as described by the David Scharp and colleagues.  Thus, stem cell therapy for T1D should not aim to use β-cell clusters in place of islet cell clusters or pseudoislets because of the need to regulate β-cell insulin secretion by the other islet cells. 
  2. With this preamble, the authors need to make changes throughout the manuscript to reflect this scientific knowledge. Thus, the use of terms, such as β-like cells, SC-β cells, are extensively used inappropriately. In most cases where "β-like cells" is used the better term should be "islet-like cells" and SC-β cells should read "SC-islet cells". 
  3. Also, while it is alright to talk about β-cell replacement, it is not so to talk about β-cell transplantation as it sends the wrong message that it is acceptable to just transplant pancreatic β-cells in patients. 
  4. On page 10, it is stated: “ In addition, islet transplantation is estimated to cost $519,000 over 20 years, whereas insulin therapy is estimated to cost $663,000 over 20 years [81].” It is necessary to specify that these costs are per patient.

Author Response

We are very grateful to Reviewer #1 for very careful and extensive review of the previous version of this manuscript and making suggestions that were helpful in the revision of the manuscript.

Comment 1) The authors must be aware of the fact that when they use the term "pseudoislet" it should refer to a cell cluster comprised of α, β, δ, PP and ε cells as described by the David Scharp and colleagues.  Thus, stem cell therapy for T1D should not aim to use β-cell clusters in place of islet cell clusters or pseudoislets because of the need to regulate β-cell insulin secretion by the other islet cells.

Response 1) Thank you for your suggestion. We add the following statement; "In vitro, several methods have been used to generate engineered pancreatic islets with a 3D structure, known as “pseudo-islets”, which are composed of α, β, δ, PP and ε cells" (On page 2, line 53-55), and we cite the paper from Dr. David Scharp (ref 19,20).

Comment 2) With this preamble, the authors need to make changes throughout the manuscript to reflect this scientific knowledge. Thus, the use of terms, such as β-like cells, SC-β cells, are extensively used inappropriately. In most cases where "β-like cells" is used the better term should be "islet-like cells" and SC-β cells should read "SC-islet cells".

Response 2) As you pointed out, we changed the terms of "β-like cells" to "islet-like cells" and "SC-β cells" to "SC-islet cells".

Comment 3) Also, while it is alright to talk about β-cell replacement, it is not so to talk about β-cell transplantation as it sends the wrong message that it is acceptable to just transplant pancreatic β-cells in patients.

Response 3) Thank you for your comment. We agree with your opinion. We revise the following five points.

<1> We add the following description in part of section 3 (On page 5, line 186-190); "In addition, β-cell replacement therapy is also expected to be a promising treatment for T1D to realize the goal of diabetes therapy that is free from insulin injection. β-cell replacement includes transplantation of pancreas, islets or SC-islet cells. In this chapter, we would like to compare transplantation of SC-islet cells and artificial pancreas (Table 1)."

<2> We also revised the statement of "β-cell replacement" to "SC-islet cells transplantation" in Table 1.

<3> "Sensor augmented insulin-pump therapy and transplantation therapy of whole pancreas or pancreatic islets may be useful strategies to compensate for β-cell function and control the blood glucose levels [4]." (On page 1、line 28-31)

<4> "Pancreas or islet transplantations are enable to regulate the blood glucose levels in T1D patients with unstable blood glucose levels [90,91]." (On page 10、line 422-424)

<5> "On the other hand, pancreas or islet transplantations has some disadvantages, such as a lack of donors for cadaveric pancreas and islets and the need to administer immuno-suppressive drugs." (On page 10、line 425-427)

Comment 4) On page 10, it is stated: “In addition, islet transplantation is estimated to cost $519,000 over 20 years, whereas insulin therapy is estimated to cost $663,000 over 20 years [81].” It is necessary to specify that these costs are per patient.

Response 4) We revised that sentence as follows (On page 10, in line 442-444). "In addition, islet transplantation is estimated to cost $519,000 per patient over 20 years, whereas insulin therapy is estimated to cost $663,000 per patient over 20 years [99]."

Reviewer 2 Report

In this review, authors compared the merits of stem cell therapy with pancreatic islet transplantation and exogenous insulin therapy. The idea of comparing and contrasting SC therapy with other therapies is well thought of. I do have some suggestions-

1) Figure 1. At the moment, the source of this image is not clear. Authors cite ref 20 for this figure. Is this image already published in the paper? Or is this an unpublished image from their lab?

2) Section 2. Authors should provide more a more comprehensive review of the papers discussed. E.g. "In a mouse study, graft rejection did not occur during the autologous transplantation of iPSCs" is lacking the information needed to draw any conclusions regarding this study.

3) Authors should start each section with a general overview followed by in-depth discussion of the most current or relevant paper in that particular field. E.g. in Section 4- it gives you the impression that having purified stem cell-islets is beneficial but then in the next paragraph authors discuss the merits of having other cells. Authors should structure the review so that it gives a clear message while at the same time discussing all the alternative possibilities.

4) Page 5-line 189. Authors should provide ref for the statement "autologous iPSCs are not thought to initiate immune response".

5) Page 5, line 224- Authors should modify the statement "....which can interrupt insulin infusion resulting in prolonged hyperglycemia that can eventually lead to diabetic ketoacidosis". 

6) Page 8, line 337- Is there any evidence that SC-beta cells may contain macrophages? 

Author Response

We would like to thank Reviewer #2 for a very careful and extensive review of previous versions of this manuscript and suggestions to help us revise the manuscript.

Comment 1) In this review, authors compared the merits of stem cell therapy with pancreatic islet transplantation and exogenous insulin therapy. The idea of comparing and contrasting SC therapy with other therapies is well thought of. I do have some suggestions-Figure 1. At the moment, the source of this image is not clear. Authors cite ref 20 for this figure. Is this image already published in the paper? Or is this an unpublished image from their lab?

Response 1) Thank you for your comment. We’re sorry for our confusing description. This image is the hPSC-derived islet organoid which we cultured in our Lab. We corrected the place of citation and described the following sentence (On page 2, in line 57-58); "Figure 1 shows the images of human pluripotent stem cells (hPSCs)-derived islet organoids."

Comment 2) Section 2. Authors should provide more a more comprehensive review of the papers discussed. E.g. "In a mouse study, graft rejection did not occur during the autologous transplantation of iPSCs" is lacking the information needed to draw any conclusions regarding this study.

Response 2) Thank you for your suggestions. The following three points have been edited in section 2.

<1> On page 2-3, in line 76-85. We referred to the possibility of the graft rejection in autologous transplantation using iPSCs. We add the following sentence. "It is thought that autologous transplantation of iPSCs does not cause graft rejection because those iPSCs are derived from patients themselves. In fact, autologous transplantation of mouse iPSCs-derived skin, bone marrow, endothelial cell, or neuronal cells and monkey iPSCs-derived neuron did not evoke immune responses [23-25]. Meanwhile, Zhao et al. reported that the autologous transplantation of mouse iPSCs-derived teratoma into subcutaneous space showed graft rejection [26]. This report indicated that the abnormal gene expression during the differentiation of iPSCs could induced T-cell-dependent immune response in autologous transplantation. An autoimmune response by autoreactive T cells for transplanted islet cells can occur in T1D recipients."

<2> On page 3, in line 100-105. We described about the immune responses in terms of HLA. The revised sentence as follows. "Human leukocyte antigen (HLA) distinguishes foreign antigens and elicits immune responses [33,34]. HLA class II is expressed on antigen-presenting cells such as dendritic cells, and present the antigen to helper T cells, resulting in initiation of antigen-specific immune responses. The class II gene shows the strongest association with T1D, and the haplotype formed by the DR and DQ genes is involved in disease susceptibility of T1D [35,36]. "

<3> On page 3, in line 113-119. We add the following sentences. "Yamaguchi et al. reported the autologous transplantation study where mouse-PSC-derived islets cells were transplanted into the kidney capsule of mice with STZ-induced diabetes [40]. In that report, SC-islets were generated by injecting mouse PSCs into Pdx-1-deficient rat blastocysts, and the SC-islets contained endothelial cells from rat origin. As a result, immunosuppressive therapy was required for the first 5 days after transplantation. Even after the withdrawal of immunosuppressive drugs, SC-islets continuously improved blood glucose levels within normal range in diabetic mice for 370 days."

Comment 3) Authors should start each section with a general overview followed by in-depth discussion of the most current or relevant paper in that particular field. E.g. in Section 4- it gives you the impression that having purified stem cell-islets is beneficial but then in the next paragraph authors discuss the merits of having other cells. Authors should structure the review so that it gives a clear message while at the same time discussing all the alternative possibilities.

Response 3) As you mentioned, the context in section 4 is difficult to understand. The revised manuscript as follows. (On page 6-7, in line 263-281). “The addition of specific factors at the correct differentiation stages results in producing functional islets from hPSCs which secrete insulin in response to glucose [59-61]. Those SC-islets, however, had a lower insulin secretion capacity, compared with native human islets, and SC-islets were still less matured (for example, they lacked the expressions of urocortin3, MAF A, and SIX3) [62]. The results of ongoing clinical trial of SC-islet transplantation for patients with T1D (NCT02239354) will provide us more information about the maintenance of graft functions [63,64]. It was reported that islet microenvironment and interactions among pancreatic endocrine cells played crucial roles in β-cell maturation, function, and survival [65]. The islet microenvironment (also referred to as the islet niche) consists of endocrine cells (α-cells, δ-cells, PP-cells, or ε-cells), pancreatic acinar cells, ductal cells, mesenchymal stromal cells, endothelial cells, resident macrophages, and extracellular matrices. Additionally, abundant islet vascular networks and interactions between islets and sympathetic or parasympathetic nerves also exist. This microenvironment is a prerequisite for re-vascularization of the grafted islets; however, the underlying mechanism remains unclear. A direct differentiation protocol from iPSCs produces not only β-cells, but also α-cells (expressing glucagon), δ-like cells (expressing somatostatin), and non-endocrine cells (which are similar to acinar cells, ductal cells, mesenchymal cells and entero-chromaffin cells) [62]. Here, we will focus on the islet microenvironment required for successful stem cell therapy for T1D.”

Comment 4) Page 5-line 189. Authors should provide ref for the statement "autologous iPSCs are not thought to initiate immune response".

Response 4) Thank you for your advice. We add reference and revise the sentence as follows (On page 5, line 211-212). "However, autologous transplantation of iPSCs might be useful for avoiding rejection because they are not thought to initiate immune responses [23,24]."

Comment 5) Page 5, line 224- Authors should modify the statement "....which can interrupt insulin infusion resulting in prolonged hyperglycemia that can eventually lead to diabetic ketoacidosis".

Response 5) Thank you for your suggestion. We revise the statement as follows (Please see on page 6, in line 247-250). "Other problems with an insulin pump include equipment trouble such as the occlusion of the insulin cannula, which can interrupt insulin infusion resulting in acute hyperglycemic complications, such as DKA and hyperosmolar hyperglycemic syndrome (HHS)."

Comment 6) Page 8, line 337- Is there any evidence that SC-beta cells may contain macrophages?

Response 6) Thank you for your comment. As you pointed out SC-beta cells should not contain macrophages. We now revised the statement as follows (Please see on page 8, line 355-359). "When SC-islet cells are transplanted, recipient MSCs and endothelial cells may change in number or in function, although this has not yet been elucidated. Interaction between transplanted SC-islet and resident macrophages may differ by the transplant site [84]. Furthermore, the effects of macrophages on SC-islet cells may be different from those of mouse islets or human islets."

Round 2

Reviewer 2 Report

Authors have revised the paper appropriately.